# Tree Communities in Three-Year-Old Post-Mining Sites Under Different Forest Restoration Techniques in the Brazilian Amazon

**Denis Conrado da Cruz** [1,*] , **José María Rey Benayas** [1] , **Gracialda Costa Ferreira** [2] and **Sabrina Santos Ribeiro** [2]

[1] Department of Life Sciences, Forest Ecology and Restoration Group, University of Alcalá, 28805 Alcalá de Henares, Madrid, Spain; josem.rey@uah.es

[2] Institute of Agricultural Sciences, Federal Rural University of Amazon, 66077830 Belém, Pará, Brazil; gracialdaf@yahoo.com.br (G.C.F.); sabrinaflorestal@hotmail.com (S.S.R.)

[*] Correspondence: conrado_denis@hotmail.com; Tel.: +34-918-854-987

**Abstract:** Forest loss and degradation in the Brazilian Amazon due to mining activities has been intense for many years. To reverse this situation, a range of restoration programs for deforested and degraded areas have been created and implemented. The aim of this study was to analyze the tree composition, successional stage, dispersal and pollination syndromes, conservation status of tree species, and proximity to seed sources under different forest restoration techniques (seedling planting, natural regeneration, and assisted natural regeneration or nucleation) implemented in post-mining sites in the Paragominas municipality (Pará, Brazil). Sixty permanent plots with a restoration age of three years were selected for tree sampling. A total of 119 species, 83 genera and 27 botanical families were identified. Sites restored with different techniques significantly differed in tree composition. Seedling planting sites exhibited the highest abundance, species richness, and diversity values. These were dominated less by pioneer species when compared to the natural regeneration and nucleation sites. Entomophilic pollination and zoochory dispersal were highly represented in the three types of restored sites. Abundance and species richness were negatively correlated with distance from plots to seed sources, and they sharply declined in natural regeneration and nucleation plots at >250 m from seed sources. Four threatened species were identified in the restored sites. We conclude that a combination of different restoration strategies at three-year-old post-mining restoration sites in the Brazilian Amazon results in the recovery of considerable levels of local tree diversity.

**Keywords:** dispersal; importance value index; pollination; seed sources

## 1. Introduction

The Brazilian Amazon rainforest contributes extensively to the world's biodiversity, as it harbors ca. 16,000 tree species. Of this total, only 227 species are dominant, representing ca. half of all trees in the region; some 4773 species are considered regulars, while 11,000 species representing only 0.12% of the trees are classified as rare [1]. Anthropogenic actions may drastically change the original forest landscape and drive many forest species to local extinctions [2–5]. Forests are suppressed or highly disturbed by mineral extraction [6]. The impacts caused by mining can reach more than 10 km [7] due to factors such as land use change, urban expansion, waste discharge, and others [8–11]. Post-mining restoration has the potential to mitigate the impacts of mining activities on tropical biodiversity [12,13], and the assessment of restoration techniques is important for the efficiency of resource use [14,15].

Forest loss and degradation spreads over 788,353 km$^2$ in the Brazilian Amazon, of which 267,393 km$^2$ (40%) are in the Pará state and 8792 km$^2$ (1%) in the Paragominas municipality, where our

study was conducted [16]. Mining has significantly contributed to such forest loss and degradation. To redress this situation, a range of restoration programs on deforested and degraded areas were created and implemented in the region [17–21]. The restoration of post-mining areas in Brazil is under the regulation of federal Law 225 (1988 Federative Constitution of Brazil) and the specific laws of each state [22]; there is, however, insufficient regulation of restoration projects and companies often fail to restore forests adequately (e.g., many restoration projects use exotic species, Cruz et al. [21]). Some results of post-mining restoration in the region have been previously published [23,24]. For instance, since the late 1970s the Mineração Rio do Norte SA company has developed a forest restoration program in Oriximiná municipality (Pará State), planting native forest species [25]. Planting a rich suite of native forest species is desirable for several reasons, including biodiversity conservation [19,26] and preventing the establishment of exotic invasive species [27]. However, commercial timber species are often used due to their lower cost [28]. Restoration projects can be expensive and inefficient [29]; thus, there is an urgent need to implement adequate techniques for successful restoration projects [30,31].

The three major techniques applied to reestablish forest vegetation in post-mining sites are seedling planting [32–34], passive restoration or natural regeneration [35,36], and assisted natural regeneration by nucleation [37,38]. All these restoration techniques are ultimately linked to secondary succession [36], i.e., they both affect and are affected by this process. Active restoration involves management techniques, such as planting, that are aimed at producing a forest with a particular composition or structure; a wide variety of approaches have been used to restore degraded forest areas [39]. Natural regeneration involves the colonization of the sites to be restored by whatever plants and animals can disperse from surrounding habitats and subsequently establish themselves; it therefore has a highly stochastic outcome [40]. Assisted natural regeneration represents an intermediate technique that involves acting in focal areas for facilitating vegetation recovery; nucleation, a type of assisted natural regeneration, pursues the establishment of woody recruits in these focal areas or nuclei to trigger forest expansion in larger areas over time through natural regeneration [41,42].

Knowledge of floristic composition is essential for all three types of restoration techniques for managing natural regeneration, selecting species to be used for restoration plantings and aiding conservation programs of threatened plant species [43–45]. The definition of functional groups of species that share similar traits has been shown to be a useful approach to understand secondary succession [46–49]. Assessing the functional composition trajectory may help us to overcome some limitations with taxonomic identification and provide more meaningful outcomes to evaluate restoration success [50]. Thus, the successional stage and the dispersal and pollination syndromes of the species involved in the restoration process, either planted or naturally established, are critical characteristics both for planning and assessing the outcomes of restoration projects [51–53]. For instance, habitat degradation disrupts key mutual interactions between animals and plants [5,54] and, consequently, affects seedling recruitment [55,56].

In this context, the present study aims to analyze the tree composition, the functional types (i.e., successional stage and the dispersal and pollination syndromes), the effect of distance between the restored sites and seed sources, and the conservation status of established tree species under different forest restoration techniques (namely seedling planting, natural regeneration, and nucleation) implemented in post-mining sites in the Paragominas municipality. Our starting hypothesis is that there are differences in tree composition and functional groups among the three forest recovery techniques (H1). We also hypothesized that the distance from seed sources and vegetation recovery at the restored sites are negatively correlated (H2). We asked: (1) Is tree diversity substantially recovered at post-mining sites in the early stages of restoration (Q1)?; (2) Is there a difference in species richness and diversity among the different restoration techniques (Q2)?; and (3) what is the conservation status of the species established under these techniques (Q3)? The answers to these questions and an assessment of the formulated hypotheses will improve the forest restoration of post-mining sites in the Brazilian Amazon and other tropical areas of the world.

## 2. Material and Methods

### 2.1. Characterization of the Study Area

The study was conducted in the Paragominas municipality (Figure 1), which is 19,465 km$^2$ in area, and has a hot and humid climate, with an annual average temperature of 26.3 °C, relative air humidity of 81%, and annual average rainfall of 1743 mm [57]. The region's soil is classified as a Yellow Dystrophic Latosol, with a very clayey texture (clay content > 700 g kg$^{-1}$). We collected data from the Degraded Area Recovery Program (PRAD) in the Hydro area of the Paragominas Mining Company S.A. (MPSA). The company owns an area of 18,668 ha, of which 4237 ha have been mined since 2006, a practice which continues in full operation today. It extracts bauxite from 12 m deep mineral layers, which results in suppressed vegetation and the removal of topsoil and soil horizons. The disturbed mine sites to be revegetated are thus the unstructured soil horizons that were removed and used later to fill in the sites where bauxite was extracted. The PRAD in this area was established in 2009 and 2339 ha had been recovered by 2019.

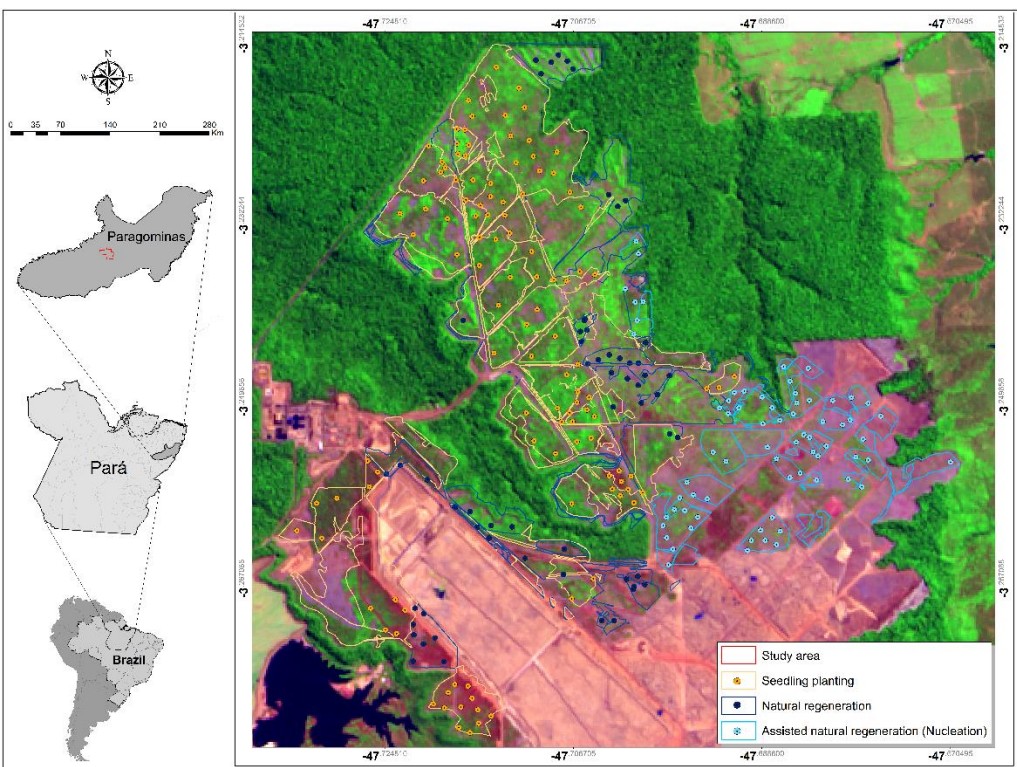

**Figure 1.** Location of the study area in Paragominas (Pará, Brazil), the three types of sites where post-mining restoration was implemented (seedling planting, natural regeneration and nucleation), and the 163 permanent plots established to assess distance to seed sources.

### 2.2. Forest Restoration Techniques

Post-mining restoration actions on the study area took place after mineral extraction finished in 2009. The first step towards forest restoration was the topographic reshaping of the ground. This stage used sub-soiling with leveling grids (trawling), driven by low-compaction tire tractors to avoid laminar erosion and water accumulation. The second step was the addition of a ca. 20-cm organic soil layer (the previously removed topsoil). This layer contained 25 m$^3$ ha$^{-1}$ of non-timber residues such as leaves and litter in the plots to be restored by nucleation. This plant material was distributed as evenly as possible to provide adequate soil protection. Finally, the techniques that were used to restore forest vegetation were (a) seedling planting, (b) natural regeneration, and (c) nucleation (Figure 2).

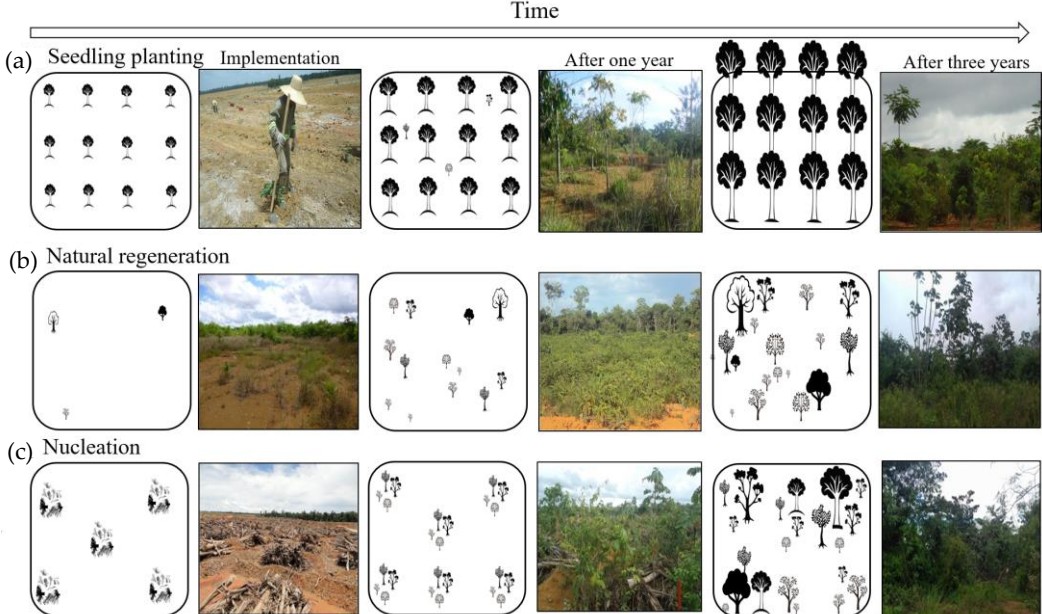

**Figure 2.** Sketch of the techniques that were used to restore forest vegetation in the studied post-mining sites and illustrative photos of the established vegetation over time. (**a**) Seedling planting, (**b**) natural regeneration, and (**c**) nucleation.

(a) Seedling plantings comprised 120 tree species that are native to the region. The seedlings were produced in the nursery of the MPSA using seedling bags (15 × 20 cm) and tubes (6 × 19 cm) filled in with organic compost (manure and black earth in the proportion of 1:3). By the time of planting, they were, on average, 30 cm tall and 4–6 months old. A total of 1111 seedlings ha$^{-1}$ were planted with a regular spacing of 3 × 3 m in pits of 0.30 × 0.30 × 0.30 m, with a 90 gr N:P:K (20:10:20) and 500 g organic compost fertilization applied to each seedling once at the time of planting. The restoration using this technique started on an area of 92.48 ha in 2009 and attained 1011 ha by 2016.

(b) Natural regeneration occurred on the added topsoil after the topographic reshaping explained above. No chemical or biological treatment was applied. It commenced on 21 ha of post-mining area in 2009 and attained 546 ha by 2016.

(c) Nucleation used nuclei of residues (leaves, branches, stumps, etc.) from the suppressed vegetation in areas (on the company's land) that would eventually be mined in the future, which were transported to the previously leveled areas. These residues were arranged in mounds of 8–10 m$^3$, which formed a "chess board" layout intended to initiate the nucleation process. This technique started in 2013 on 9.16 ha and was later extended in 2014, 2015 and 2016, up to a total of 81.63 ha.

*2.3. Data Collection*

A total of 163 permanent plots were established in the restored areas from 2009 to 2016, 83 at the seedling planting sites, 56 at the natural regeneration sites, and 24 at the nucleation sites (Table S1). They were 20 × 50 m (1000 m$^2$) for the seedling planting and nucleation sites and 10 × 25 m (250 m$^2$) for the natural regeneration sites. All trees with diameter at breast height (dbh) ≥ 15 cm were surveyed for eight consecutive years between 2009 and 2016. We acknowledge that, by sampling only trees with dbh > 15 cm, many smaller trees were missed which undoubtedly contribute to the diversity in the studied sites, while the obtained results refer to only the most successfully fast-growing species. However, we only analyzed a subset of 60 plots with a restoration age of 3 years that were randomly selected (see below), namely 10 plots at the seedling planting sites, 10 at the nucleation sites, and 40 at the natural regeneration sites; the number of plots at the nucleation sites was higher due to their smaller size, so the total area sampled was identical for the three restoration techniques.

To analyze tree composition at the restored sites, we calculated Importance Value Indices (IVI) of both species and families, which are based on relative frequency (number of plots where taxa occurred), relative density (number of individuals) and relative dominance (basal area across plots) (Table S2). The tree diversity was analyzed using the Shannon–Weaver, Simpson and Pielou Equability indices (Table S2).

The surveyed species list was classified into groups based on the successional stage, labeled as pioneer (P), initial secondary (IS), late secondary (LS), and climax (C) species [58]. The IVI was also calculated for these successional stage groups. We adopted the morphological criteria defined by Pijl [59] to classify the species into the following types of dispersal syndrome: anemochory, autochory, barochory, hydrochory, and zoochory. The type of pollination was classified as anemophilic, entomophilic, melittophilic, chiropterophilic, and zoophilic according to Real [60] (Tables S3–S5).

We measured the distance between each restored plot and the nearest seed source. For this, an image of the Landsat-8 satellite (available at https://glovis.usgs.gov/app/, accessed on 7 July 2018 with orbit/dot-223/62) was classified according to an unsupervised method to distinguish and delineate forest remnants around the restored sites. Then, the minimum distance of the plot to the nearest vegetation point was defined by the Euclidean distance method with ArcGIS 10.1 [61]. The distance between the plots and the nearest seed source was classified into three categories, namely short (<250 m), intermediate (251–500 m) and long (> 501 m) distance.

Finally, the species were classified according to their conservation status or risk of extinction using the National Center for Flora Conservation (CNCFLORA; http://cncflora.jbrj.gov.br/portal/), which follows the guidelines, categories and criteria established by the International Union for Conservation of Nature (IUCN; https://www.iucnredlist.org/, accessed on 25 November 2019). The nine groups were: Least Concern (LC), Near Threatened (NT), Vulnerable (VU), Endangered (EN), Critically Endangered (CR), Extinct in the Wild (EW), Date Deficient (DD), and Not Evaluated (NE).

### 2.4. Data Analysis

A resampling technique was used to randomly select the 60 analyzed plots and ensure identical and comparable sampling areas among the different restoration types. This selection was repeated 1000 times, and the tree abundance, IVI and diversity indices were calculated on each occasion. Finally, we calculated the average and standard error of abundance, IVI and diversity indices with an alpha of 0.05 across all repetitions.

Permutational Analysis of Variance (PERMANOVA, [62]) was used to test differences in tree composition, successional stage, and dispersal and pollination syndromes among sites restored with different techniques, with the R [63] vegan package [64]. A multidimensional scaling ordination (NMDS) [65] was used to visualize similarities among plots in terms of these characteristics, using the R version 3.6.3 MASS package [66]. The diversity indices were also calculated with the R vegan package. Finally, we correlated abundance and species richness and the distance of each plot to the closest patch of remnant forest.

## 3. Results

### 3.1. Tree Composition at the Restored Sites

We observed 767 individual trees from 119 species, 83 genera and 27 botanical families within the 60 plots (3 ha in total) analyzed in this study. Of this total, 526 trees representing 101 species, 73 genera and 20 families occurred at seedling planting sites (Table S3), 155 trees, 14 species, 11 genera and 10 families at natural regeneration sites (Table S4) and 86 trees, 13 species, 10 genera and eight families at nucleation sites (Table S5). Seedling planting plots exhibited the highest tree abundance (Table 1).

The species with highest IVI at seedling planting sites were *Protium* sp. (7.1% ± 0.16), *Inga alba* (Sw) Willd. (4.7% ± 0.07) and *Khaya ivorensis* A Chev. (4.6% ± 0.11) (Figure 3a); at natural regeneration sites they were *Croton matourensis* Aubl. (50.2% ± 0.19), *Solanum crinitum* Lam. (13.5% ± 0.1) and *Vismia*

*guianensis* (Aubl.) Choisy. (10.2% ± 0.09) (Figure 3b); and at nucleation sites they were *Solanum crinitum* Lam. (25.7% ± 0.18), *Cecropia* sp.1 (21.4% ± 0.39) and *Cecropia distachya* Huber (19.1% ± 0.2) (Figure 3c). The families with highest IVI were Malpighiaceae, Euphorbiaceae and Urticaceae at each restoration type, respectively (Figure S1).

**Table 1.** Species richness and diversity indices (mean ± se) at post-mining restored sites. Different letter superscripts indicate statistically significant (p < 0.05) values between sites according to Tukey's test.

| Restoration Technique | Abundance | Richness | Diversity Indices | | |
|---|---|---|---|---|---|
| | | | Shannon | Simpson | Pielou |
| Seedling planting | 525 ± 2.71 [a] | 59.5 ± 0.36 [a] | 3.5 ± 0.01[a] | 0.96 ± 0 [a] | 0.87 ± 0 [a] |
| Natural regeneration | 154 ± 0.6 [b] | 13.5 ± 0.04 [b] | 1.5 ± 0.01 [b] | 0.64 ± 0 [b] | 0.59 ± 0 [b] |
| Nucleation | 85.7 ± 0.81 [b] | 11.3 ± 0.15 [b] | 1.9 ± 0.01 [c] | 0.81 ± 0 [b] | 0.8 ± 0 [b] |

PERMANOVA showed that sites restored with different techniques significantly differed in their tree composition (Table 2a). Further, seedling planting sites were clearly segregated from the two other sites and the natural regeneration sites exhibited the highest variability, according to the NMDS (Figure 4a).

**Table 2.** PERMANOVA of (a) tree species composition, (b) successional status and (c) dispersal and (d) pollination syndromes among post-mining restoration types.

| (a) Tree Species Composition | | | | | |
|---|---|---|---|---|---|
| | df | SS | MS | F.Model | R2 | Pr(>F) |
| Techniques | 2 | 8.582 | 4.2909 | 11.808 | 0.1926 | 0.001*** |
| Residuals | 99 | 35.976 | 0.3634 | | 0.8074 | |
| Total | 101 | 44.558 | | | 1.0000 | |

| (b) Successional Status | | | | | |
|---|---|---|---|---|---|
| | Df | SS | MS | F.Model | R2 | Pr(>F) |
| Techniques | 2 | 12.851 | 6.4254 | 47.329 | 0.48879 | 0.001*** |
| Residuals | 99 | 13.440 | 0.1358 | | 0.51121 | |
| Total | 101 | 26.291 | | | 1.00000 | |

| (c) Dispersal Syndrome | | | | | |
|---|---|---|---|---|---|
| | df | SS | MS | F.Model | R2 | Pr(>F) |
| Techniques | 2 | 12.732 | 6.3658 | 32.195 | 0.39409 | 0.001 *** |
| Residuals | 99 | 19.575 | 0.1977 | | 0.60591 | |
| Total | 101 | 32.306 | | | 1.00000 | |

| (d) Pollination Syndrome | | | | | |
|---|---|---|---|---|---|
| | df | SS | MS | F.Model | R2 | Pr(>F) |
| Techniques | 2 | 16.085 | 8.0423 | 48.965 | 0.49728 | 0.001 *** |
| Residuals | 99 | 16.260 | 0.1642 | | 0.50272 | |
| Total | 101 | 32.345 | | | 1.00000 | |

Significance codes: '***' 0.001.

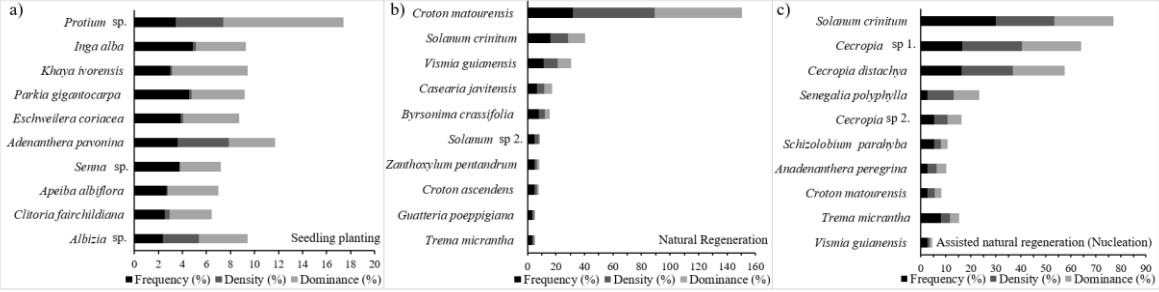

**Figure 3.** Relative frequency, density and dominance of the species with highest Importance Value Indices (IVI) (i.e., the sum of these three components) at the seedling planting (**a**), natural regeneration (**b**) and nucleation (**c**) sites.

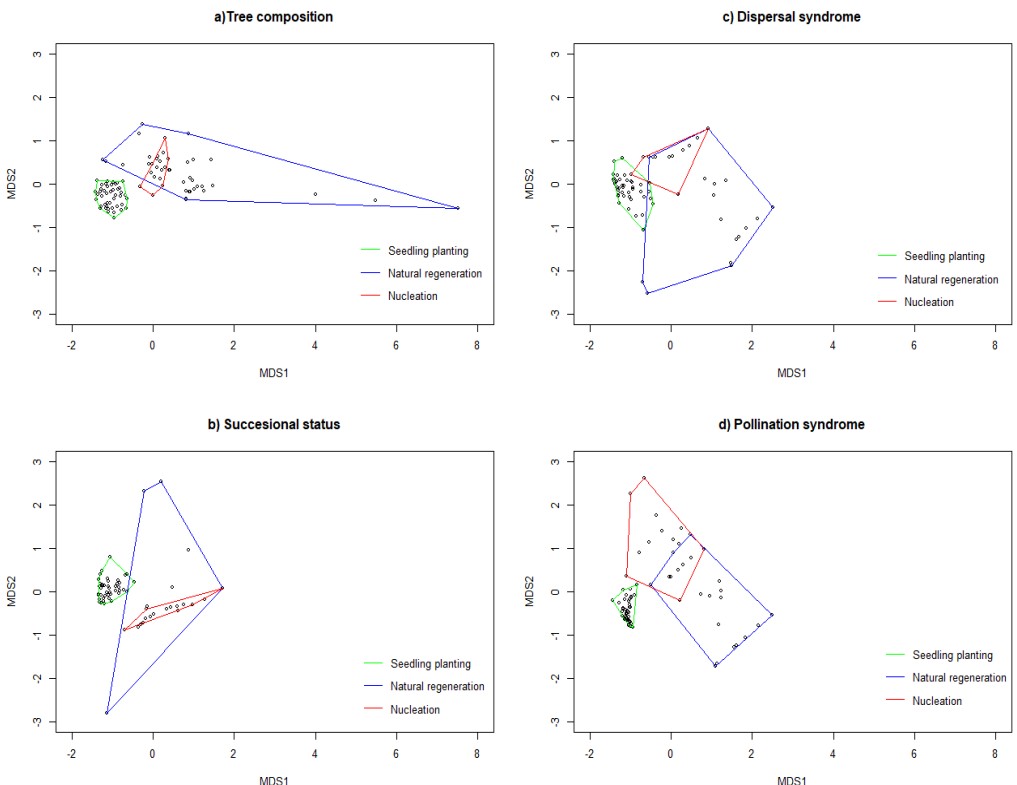

**Figure 4.** Plot ordination of (**a**) tree composition, (**b**) successional status, (**c**) dispersal syndrome, and (**d**) pollination syndrome under seedling planting, natural regeneration and nucleation techniques according to a NMDS.

### 3.2. Species Richness and Diversity

Seedling planting sites exhibited the highest species richness, Shannon, Simpson, and Pielou index values, whereas the diversity values found in nucleation sites were slightly greater than in natural regeneration sites (Table 1).

### 3.3. Functional Types

PERMANOVA showed that successional status, dispersal and pollination syndromes significantly differed among sites restored with different techniques (Table 2b–d). Again, seedling planting sites were clearly segregated from the two other sites and the natural regeneration sites were the most widely spread on the NMDS plot (Figure 4b–d).

The majority of species identified were categorized in the pioneer group (41), followed by the initial secondary (37), late secondary (39), and climax (2) groups. Nucleation (92% ± 0.2%) and natural regeneration (87% ± 0.1%) sites showed higher pioneer IVI than seedling planting sites (30% ± 0.2%) (Table S6). Initial secondary (25% ± 0.2%) and late secondary (42% ± 0.2%) species were the most prominent in the seedling planting sites and of marginal importance in the natural regeneration (6% ± 0.1% and 7% ± 0.1%, respectively) and nucleation (5% ± 0.2% and 2% ± 0.2%, respectively) sites.

The surveyed species were classified as zoocoric (56%), autocoric (21%), anemocoric (17%), barocoric (4%), and hydrocoric (2%) according to their dispersal syndrome. The three types of restoration sites were dominated by zooric species followed by autocoric species, whereas anemocoric species were also relevant in the seedling planting sites (Figure 5a).

Most species were classified as entomophilic (72%), which were followed by meliophilic (19%), anemophilic (5%), zoophilic (3%), and chiropterophilic (1%) species. The entomophilic syndrome dominated the three restoration techniques, and the meliophilic and anemophilic syndromes were also relevant at natural regeneration and nucleation sites, respectively (Figure 5b).

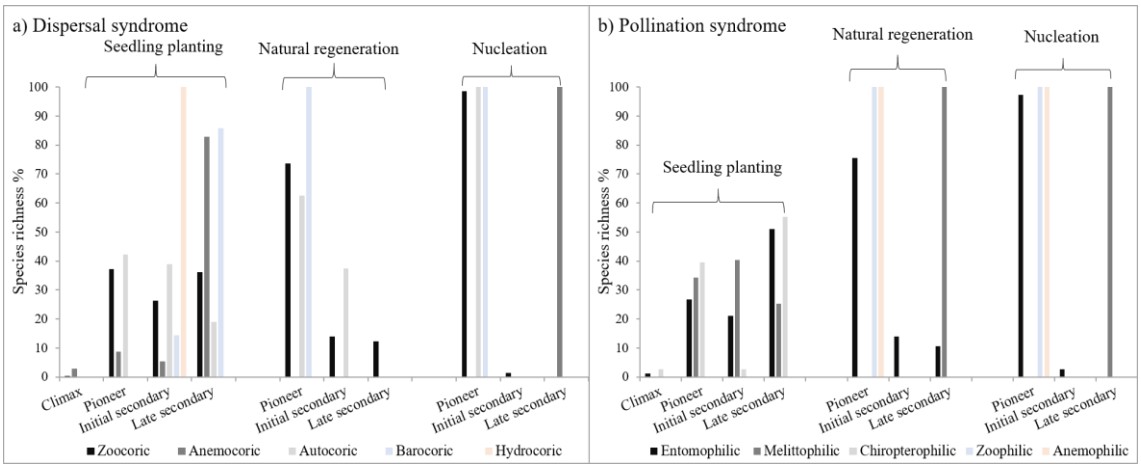

**Figure 5.** Proportion of species according to the (**a**) dispersal and (**b**) pollination syndromes in seedling planting, natural regeneration and nucleation sites.

### 3.4. Effects of Distance to Seed Sources

Tree abundance and species richness at the restored plots were negatively correlated with distance to the seed sources under the three restoration techniques (Figure 6). The seedling planting plots exhibited the highest correlation coefficients and the nucleation plots the lowest. Noticeably, abundance and species richness gradually declined with their distance to forest edge at the seedling planting sites. However, both nucleation and natural regeneration sites exhibited a sharp decline in the abundance and species richness along the shortest distances (<250 m) from seed sources; abundance and richness were very low and remained constant at distances > 250 m.

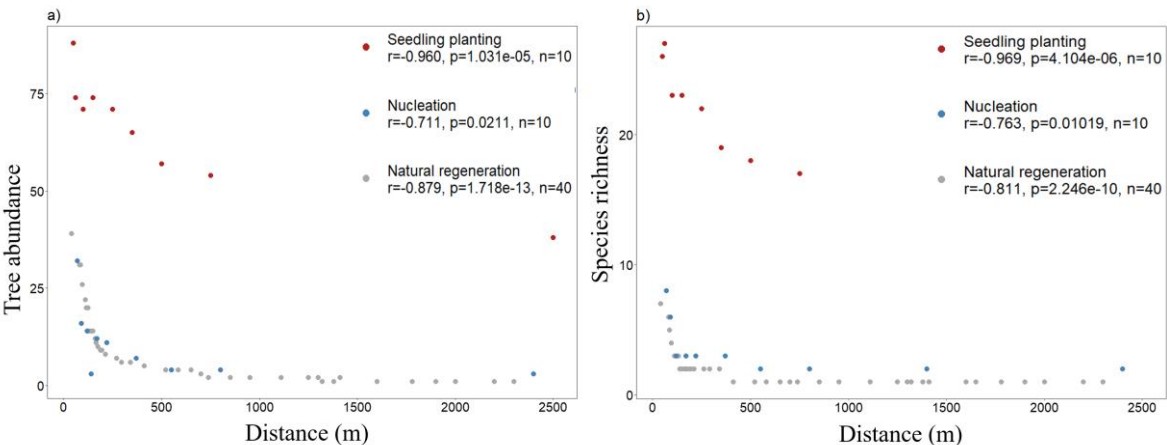

**Figure 6.** Tree abundance (**a**) and species richness (**b**) in the assessed plots at post-mining restored sites located at different distances from seed sources.

The majority of tree species that were established close (<250 m) to seed sources were mostly of zoochory dispersal (45% in seedling planting, 93% in natural regeneration, 69% in nucleation plots) and entomophilic pollination (87% in seedling planting; 64% in natural regeneration; 62% in nucleation plots). Regarding the successional status, 44% of the species in the seedling planting sites were late secondary, whereas 79% and 77% were pioneers in the natural regeneration and nucleation sites, respectively.

### 3.5. Conservation Status

Only 11 species found in this study have been evaluated by the National Center for Plant Conservation and all of them were planted. Four species were threatened and assigned to the categories

VU (*Swietenia macrophylla* King., *Hymenaea parvifolia* Huber and *Cedrela fissilis* Vel. L) and EN (*Vouacapoua americana* Aubl.). The LC category was represented by *Pterocarpus santalinoides* L'Hér. Ex DC., *Bowdichia nitida* Spruce ex Benth., *Lecythis lurida* (Miers) S.A. Mori., *Hymenaea courbaril* L., and *Genipa americana* L., the NT category by *Handroanthus impetiginosus* (Mart. Ex DC.) Mattos., and the DD category by *Ficus malacocarpa* Standl.

## 4. Discussion

This study assessed the outcomes of contrasting forest restoration techniques in terms of tree composition, functional types, and conservation value of the established tree species in Amazonian three-year-old post-mining sites. Overall, our results support our starting hypothesis on the differences in tree composition and functional groups among the three forest recovery techniques (H1). We also found an acceleration of vegetation recovery for sites close to seed sources (H2), a substantial level of early established tree species (Q1), differences in species richness and diversity among the different restoration techniques (Q2), and a few endangered tree species established in the young restored sites (Q3).

### 4.1. Effects of Restoration Techniques on Tree Community

We found that sites restored with different techniques differed in tree composition. The success in establishing a forest restoration project depends, among other factors, on the selection of the techniques to be implemented [29,31,40,67]. This selection should consider context factors such as the level of initial degradation, the characteristics of the landscape and the reference community that is the subject of restoration [67–70]. A current debate is to what extent it is justified to use active restoration techniques such as seedling planting over passive restoration, as the former is much more expensive than the latter [36,50]. This debate has been fueled by recent global meta-analyses related to forest restoration, which have found that recovery levels of biodiversity, forest structure and function indicators are often greater, or at least similar, for passive restoration than for active restoration in the long term [29,71,72]. However, in the current study, seedling planting sites exhibited the highest abundance, species richness, and diversity values. This finding can be attributed to the high initial degradation level of post-mining sites, in accordance with Reid et al. [73] who found a positive site selection bias in meta-analyses comparing natural regeneration to active forest restoration (i.e., passive restoration outperforms or is similar to than active restoration because passively restored sites tend to be less disturbed than those sites restored by active restoration techniques).

The key constraints for vegetation recovery are: (1) seed bank availability and germination [74]; (2) dispersal limitation, because seed sources in restoration sites include remote and dispersal vectors may be rare [75]; (3) abiotic limitation, such as low water availability, extreme temperatures, poor soil structure, and low nutrient availability [76–78]; and (4) biotic limitations, such as competition from herbaceous vegetation and herbivory [79,80]. In this study, community composition and the features of taxa at the three types of restored sites indicate that seedling planting mitigates the constraints for natural regeneration. Thus, the dominant families and species with highest IVI, overall tree composition, and most relevant functional types according to the successional status were notably different in the seedling planting sites, which were less dominated by pioneer species compared to the natural regeneration and nucleation sites. However, the dispersal and pollination syndromes mostly overlapped across all restoration types as, in the tropics, most dispersal is zoochory [81] and most pollination is carried out by insects [82,83]. The dispersal of plots in our NMDS analyses reflects the high stochasticity of restoration outcomes following natural regeneration [84,85] and the more predictable outcomes of seedling planting [86,87].

Not surprisingly, the most abundant and species-rich family in our study was Fabaceae, which has high ecological plasticity and a high capacity of nitrogen fixation and facilitates the establishment of other species [88]. Gama et al. [89] showed that *Guatteria poeppigiana* Mart. and *Manihot brachyloba* Müll. Arg., which were identified in the natural regeneration sites of this study, have a high potential

for restoring degraded areas. In the nucleation sites, on top of the natural recovery progress, the nuclei create microhabitats that facilitate the entry of several plant species into the system [42,90]. Species such as *Cecropia distachya* Huber and *Solanum crinitum* Lam, which were present in the nucleation sites, are also considered as important for the recovery of degraded areas [91].

### 4.2. Proximity to Seed Sources Accelerates Vegetation Recovery

We found that both abundance and species richness and distance to seed sources were negatively correlated across all restored sites, as Cubiña [92] also reported, meaning that proximity to seed sources accelerates vegetation recovery. This well-known pattern [93,94] supports the necessity of conserving forest remnants within and around mining projects to assure a high seed pressure [23,95,96], and the utility of planting tree islands that eventually may trigger applied nucleation [42,97]. In central Amazonia, abundance and species richness have been found to be affected at distances between 10 to 400 m from the fragment edges [98].

Our results detected that natural regeneration is severely limited at distances > 250 m in the natural regeneration and nucleation sites, in agreement with other studies [94,99,100]. Some studies have shown that geographical distance influences the distribution of plants in the tropics, and that the variation in richness and species composition in a given region is limited by seed dispersal [101,102]. At restored sites in the proximities of forest edges, there may be a higher abundance of pioneers and initial secondary species established from the seed bank contained in the topsoil [103].

### 4.3. Relevance for Biodiversity Conservation

Our study provides further evidence that restoration is a tool for biodiversity conservation [68,104,105]. Thus, a total of 119 species, 83 genera and 27 botanical families were identified in 30 ha of a three-year-old restored mined area. Similarly, Rankin-de-Merona et al. [106] found 53 botanical families in a survey of 70 ha in the Amazonian rainforest close to Manaus. Other studies have reported a substantial recovery of biodiversity in post-mining restoration sites around the world [80,107–112]. Furthermore, we found the quick establishment of threatened tree species. This is of particular importance in biodiversity hotspots such as the Amazonian rainforest, where it is estimated that between 5% and 9% of all species will be threatened with extinction by 2050 [2].

## 5. Conclusions

Our study has shown that (1) a substantial amount of local tree diversity was recovered in young post-mining restored sites in the Brazilian Amazon rainforest (119 species, 83 genera and 27 botanical families). However, (2) sites restored with different techniques showed significantly different tree composition (e.g., they were dominated by Fabaceae at seedling plantings, Euphorbiaceae at natural regeneration sites, and Urticaceae at nucleation sites). Furthermore, the restoration sites also (3) differed in their abundance, species richness, and diversity values, which were highest at the seedling planting sites, and functional groups based on the successional status, dispersal and pollination syndromes. Thus, seedling planting sites were dominated less by pioneer species compared to the natural regeneration and nucleation sites, whereas entomophilic pollination and zoochory dispersal were highly represented at the three types of restored sites. Abundance and species richness (5) were negatively correlated with distance of plots to seed sources. However, whereas these values sharply declined in natural regeneration and nucleation plots at > 250 m from seed sources, they were more constant in seedling planting sites. Noticeably, (6) four threatened species were identified in the restored sites. Overall, the combination of different restoration strategies at three-year-old post-mining restoration sites resulted in considerable levels of recovery of local tree diversity.

**Supplementary Materials:** The following are available online at http://www.mdpi.com/1999-4907/11/5/527/s1, Figure S1: Mean Importance Value Indices for each botanical family found at seedling planting, natural regeneration, and nucleation sites., Table S1: Implementation of forest restoration plots, Table S2: Phytosociological analysis of vegetation structure, Table S3: Species list at the seedling planting sites with phytosociological values, dispersal

and pollination syndromes, and successional group (PI = Pioneers species; IS = Initial secondary species; LS = Late secondary and CL = Climax species), Table S4: Species list at the natural regeneration sites with phytosociological values, dispersal and pollination syndromes, and successional group (PI = Pioneers species; IS = Initial secondary species; LS = Late secondary and CL = Climax species), Table S5: Species list at the nucleation sites with phytosociological values, dispersal and pollination syndromes, and successional group (PI = Pioneers species; IS = Initial secondary species; LS = Late secondary and CL = Climax species), Table S6: Phytosociological analyses of the ecological groups in the seedling planting, natural regeneration and nucleation sites.

**Author Contributions:** D.C.d.C., G.C.F. and S.S.R. conceived and designed the experiments; D.C.d.C., G.C.F. and S.S.R. performed the experiments; D.C.d.C., J.M.R.B., G.C.F. and S.S.R. analyzed the data; D.C.d.C. and J.M.R.B. wrote the paper. All authors have read and agreed to the published version of the manuscript.

**Funding:** This research was funded by Hydro through the Biodiversity Research Consortium Brazil–Norway (BRC) grant number BRC0010.

**Acknowledgments:** The authors are grateful to Federal Rural University of Amazon—UFRA, Federal University of Pará—UFPA, Museum Emílio Goeld – MPEG, University of Oslo, CNPq—National Council for Scientific and Technological Development of Brazil (203159/2014-4/GDE), and the REMEDINAL program (TE-CM S2018/EMT-4338) of the Madrid Autonomous Government. The authors also acknowledge the input from two anonymous reviewers who improved the contents and presentation of this manuscript.

**Conflicts of Interest:** The authors declare no conflict of interest. The founding sponsors had no role in the design of the study; in the collection, analyses, or interpretation of data; in the writing of the manuscript, and in the decision to publish the results.

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
