# Peer review of "Tree Communities in Three-Year-Old Post-Mining Sites Under Different Forest Restoration Techniques in the Brazilian Amazon"

_forests, doi:10.3390/f11050527_

Round 1

Reviewer 1 Report

This paper compares three ecosystem restoration methods after surface mining in the State of Para, Brazil: planting: tree planting (some 120 spp.!), unaided natural recovery, and the use of nucleation sites or assisted natural regeneration. The research is well described, the sampling and analysis seems appropriate, and the results are certainly interesting and useful.

I have several major concerns that should be appropriately addressed before this manuscripts should be published, however.

  • On page 2, lines 1 and 2: these areas supposedly affected by mining are huge and the numbers should be checked. For example, 8792 km2 would be some 45% of the Paragominas municipality. Should these areas instead perhaps be hectares (ha), or has there been an error in conversion?
  • Greater background on the mining operations and resulting substrate for revegetation is needed. What mineral was mined from these sites? Is the resulting surface unconsolidated overburden, fractured rock, or post-processing tailings? What is its texture, coarse fragment content, and depth? Are there trace metals or other soil toxicities (e.g., Al concentrations) that are constraints to plant growth?
  • More details on the reclamation processes are needed in order to repeat and interpret them. From where were the organic materials (no mineral soil?) obtained that were spread? Check calculations: 25m3/ha would result in a covering only 2.5 cm thick! How big or how old were the seedlings that were planted? Were they container-grown (if so, what volume of containers, in what medium?) or bare root? What fertilizer was applied and how much?
  • The third treatment is variously described as “nucleation” and “assisted natural regeneration”. I suggest just using nucleation throughout to avoid confusion, though you may wish to clearly define this treatment as a form of assisted natural regeneration when first described.
  • The terms dispersion and dispersal are used synonymously throughout, when usually dispersal (movement of seeds) is being described. Dispersion (which denotes spatial distribution) is appropriate when describing the spread of points in the ordination biplots.
  • Constraints of this sampling approach and analysis need to be acknowledged. In particular, by sampling only trees with DBH >15cm, many smaller trees are missed which undoubtedly contribute to the diversity, while these results refer to only the most successfully, fasted growing species.

The writing and English grammar is generally good, as is the presentation of data. Nonetheless, I provide some minor line by line suggestions and corrections below.

Page 2, Lines 1 & 2: confirm area numbers; perhaps these should be in ha?

  1. 2., L. 9: 1970s … company has developed a …
  2. 2, L. 11: In spite of planting a rich suite…
  3. 2, 3rd paragraph, L. 1: “A knowledge of floristic …”
  4. 2, 3rd para, L. 7: dispersal [not dispersion, which denotes spatial pattern]
  5. 2, 4th para, L. 2: dispersal
  6. 2, 4th para., 2nd-last line: delete “provide light to”
  7. 3, L. 1: “…which is 19,465 km2 in area, and has a hot …”
  8. 3, L. 5: continues [not continuous]

   - what was mined? What was the nature of the minespoils, waste rock or overburden to be revegetated here?

Section 2.2, L5: presumably this should be 25 m3 not 25 m2; then check your numbers, as 25 m3/10,000m2 leaves a thickness of only 0.0025m = 2.5 cm

  • From where were these organic materials obtained? Was there no mineral soil with them?

Figure 2: is the x-axis time? Please label, providing years or range of years if possible [from the paper’s title, I assume these are years 1, 2 and 3]

  1. 4, section a) how big were the seedlings? Were they container grown or bare root? If container grown, what volume and in what medium (which was then presumably transferred with them)? What fertilizer formula was applied, at what rate (per planting hole and translated to per ha).
  2. 4., Section 2.3, L. 4: confirm that you only measured trees with DBH > 15 cm for the Year 3 sampling. This suggests that there were probably many additional (smaller trees) found on these sites which are not included in the analysis, which really emphasizes only the most successful ones. This needs to be emphasized.
  3. 5, Lines 1 & 2: from where were these successional stage labels derived? [you quote sources for dispersal and pollination syndromes; please do the same here]
  4. 5, 2nd para., L. 2; also 3rd para., L. 4: match the font of URLs with the rest of the text
  5. 5, Results [not Result]

Section 3.1, L. 1: Instead of “surveyed” maybe say “observed” or “detected” or “identified”

  1. 6, L. 1 & L. 5 under the Table: Protium sp. and Cecropia sp. should have sp. without italics

- same correction needed in Figure 3 and anywhere else in the manuscript

  1. 6, last 2 lines: rephrase, dropping “were the most dispersed, i.e.”
  2. 7, Section 3.3., L. 1: dispersal [not dispersion]
  3. 7, Sect. 3.3, L. 4: “widely spread” [so as not to confuse with seed dispersal syndrome]
  4. 8, Fig. 4, Panel c): Dispersal syndrome
  5. 8, Lines 2-6 under Fig. 4: please match the number of decimal places in the means to that used for standard error (S.E.)
  6. 8, L. 8: dispersal syndrome
  7. 9, Fig. 5: dispersal syndrome [both in the caption and in the panel a) label]
  8. 10, L. 2: re: diversity in the planted treatment sites – were these spp. volunteers on the planting sites, or are you referring solely to those planted? Or (ideally), can you provide the contribution of each origin?
  9. 10, Section 4.1, L. 5: rephrase to “…to what extent it pays to use active …”
  10. 10, 2nd-last line of first paragraph in Section 4.1: “… or is similar to …”
  11. 10, last line of first para. In Sect. 4.1: “…sites tend to be less disturbed”
  12. 10, 2nd paragraph in Sect. 4, L. 4: biotic limitations [not limitation]

            - Line 6: “indicates” [rather than “reflects”]

            - Lines 10 & 11: “dispersal” [not dispersion, both instances]

  1. 11, L. 1: please hyphenate “species-rich family”
  2. 11, Section 4.2, first para., L. 6: delete “the” before “central”
  3. 11, Section 5 heading: Conclusions [not Conclusion – you clearly have several conclusions!]
  4. 5, Line 8 under Conclusions: dispersal

Author Response

May 1st 2020

Dr. Timothy A. Martin, Editor-in-Chief

Ms. Brenda Zhao, Assistant Editor

Forests

Dear Dr. Timothy and Brenda,

Please find attached a revised version of our ms forests-783506 entitled “Tree communities in 3-yr-old post-mining sites under different forest restoration techniques in the Brazilian Amazon”, which was found suitable for publication at Forests after minor revisions.

We provide below a point-by-point response letter to the comments from two anonymous reviewers (“in italics”, our responses in regular style preceded by R.-). These comments have greatly improved both the contents and presentation of the ms.

We attach two “versions” of the revised ms. One highlights the major changes in red (the […]_HC file) and the other one is a clean version. Please note that the line numbers by our responses to the reviewer’s comments refer to the revised clean version.

We are looking forward to your news. Sincerely

Signed by Denis Conrado in behalf of all co-authors

Response to Reviewer #1

Note: the line numbers by our responses to the comments refer to the clean revised version.

General comments

 “This paper compares three ecosystem restoration methods after surface mining in the State of Para, Brazil: planting: tree planting (some 120 spp.!), unaided natural recovery, and the use of nucleation sites or assisted natural regeneration. The research is well described, the sampling and analysis seems appropriate, and the results are certainly interesting and useful”.

“I have several major concerns that should be appropriately addressed before this manuscript should be published, however.”

R.- We appreciate the general positive comment about our ms.

"On page 2, lines 1 and 2: these areas supposedly affected by mining are huge and the numbers should be checked. For example, 8792 km2 would be some 45% of the Paragominas municipality. Should these areas instead perhaps be hectares (ha) or has there been an error in conversion?".

R.- The area values are correct, Paragominas municipality is a 19,465 km2 territory, and ca. 8,792 km2 (45%) of this total was loss or degraded. Mining largely contributes to this forest loss and degradation, but other activities and land use types contribute to it (agriculture and livestock, for example). We have rephrased this statement as follows in the interest of clarity: “Forest loss and degradation spreads over 788,353 km2 in the Brazilian Amazon, of which 267,393 km2 (40%) are in the Pará state and 8,792 km2 (1%) in the Paragominas Municipality where our study was conducted [16]. Mining has significantly contributed to such forest loss and degradation”.

"Greater background on the mining operations and resulting substrate for revegetation is needed. What mineral was mined from these sites? Is the resulting surface unconsolidated overburden, fractured rock, or post-processing tailings? What is its texture, coarse fragment content, and depth? Are there trace metals or other soil toxicities (e.g., Al concentrations) that are constraints to plant growth?".

R.- We have added the information suggested by the reviewer in the following statements: “The region's soil is classified as a Yellow Dystrophic Latosol, with a very clayey texture (clay content > 700 g kg-1). We collected data from the Degraded Area Recovery Program (PRAD) in the Hydro area of the Paragominas Mining Company S.A. (MPSA). The company owns an area of 18,668 ha of which 4,237 ha have been mined since 2006 and continues in full operation today. It extracts bauxite from 12 m deep mineral layers that results in suppressed vegetation and removal of topsoil and soil horizons. The disturbed mine sites to be revegetated are thus the unstructured soil horizons that are used to fill in the sites where bauxite was extracted” (lines 98-105).

" More details on the reclamation processes are needed in order to repeat and interpret them. From where were the organic materials (no mineral soil?) obtained that were spread? Check calculations: 25m3/ha would result in a covering only 2.5 cm thick! How big or how old were the seedlings that were planted? Were they container-grown (if so, what volume of containers, in what medium?) or bare root? What fertilizer was applied and how much?".

R.- We double checked the 25 m3/ha calculation and it is correct (line 116). The information suggested by the reviewer has been added as follows: “(a) Seedling plantings comprised 120 tree species that are native to the region. The seedlings were produced in the nursery of the MPSA using seedling bags (15 x 20 cm) and tubes (6 x 19 cm) filled in with organic compost (manure and black earth in the proportion of 1:3). By the time of planting, they were on average 30 cm tall and 4-6 months old. A total of 1,111 seedlings ha−1 were planted on a regular spacing of 3 × 3 m in pits of 0.30 x 0.30 x 0.30 m, with a 90 gr N:P:K (20:10:20) and 500 g organic compost fertilization applied to each seedling once at the time of planting. The restoration using this technique started on an area of 92.48 ha in 2009 and attained 1,011 ha by 2016” (lines 125-131).

“The third treatment is variously described as “nucleation” and “assisted natural regeneration”. I suggest just using nucleation throughout to avoid confusion, though you may wish to clearly define this treatment as a form of assisted natural regeneration when first described”.

R.- We followed the reviewer´s suggestion (see line 66 and others throughout the text).

The terms dispersion and dispersal are used synonymously throughout, when usually dispersal (movement of seeds) is being described. Dispersion (which denotes spatial distribution) is appropriate when describing the spread of points in the ordination biplots”.

R.- We consistently use dispersal throughout the text.

Constraints of this sampling approach and analysis need to be acknowledged. In particular, by sampling only trees with DBH >15cm, many smaller trees are missed which undoubtedly contribute to the diversity, while these results refer to only the most successfully, fasted growing species”.

R.- We have literally added the reviewer’s comment to the text (lines 144-147).

Minor comments

  1. Page 2, Lines 1 & 2: confirm area numbers; perhaps these should be in ha?

R.- This comment was responded above.

  1. Page 2., L. 9: 1970s … company has developed a …

R.- We did so (line 52).

  1. Page 2, L. 11: In spite of planting a rich suite….

R.- We eliminated “In spite of” following the suggestion of Reviewer 2 (line 53).

  1. Page 2, 3rd paragraph, L. 1: “A knowledge of floristic …”.

R.- We did so (line 70).

  1. Page 2, 3rd para, L. 7: dispersal [not dispersion, which denotes spatial pattern].

R.- We did so (line 76).

  1. Page 2, 4th para, L. 2: dispersal.

R.- We did so (line 82).

  1. Page 2, 4th para., 2nd-last line: delete “provide light to”.

R.- We did so (line 92).

  1. Page 3, L. 1: “…which is 19,465 km2 in area, and has a hot …”

R.- We did so (line 96).

  1. Page 3, L. 5: continues [not continuous].

R.- We did so (line 102).

  1. what was mined? What was the nature of the minespoils, waste rock or overburden to be revegetated here?

R.- Please see the response to a major comment above.

  1. Section 2.2, L5: presumably this should be 25 m3 not 25 m2; then check your numbers, as 25 m3/10,000m2 leaves a thickness of only 0.0025m = 2.5 cm.

R.- We corrected the typo (Line 116); please see the response to a major comment above

  1. From where were these organic materials obtained? Was there no mineral soil with them?

R.- We completed the information of the sentence as follows: “Nucleation used nuclei of residues (leaves, branches, stumps, etc.) from the suppressed vegetation on areas that would eventually be mined in the future in the company’s land, which were transported to the previously leveled areas” (lines 135-137).

  1. Figure 2: is the x-axis time? Please label, providing years or range of years if possible [from the paper’s title, I assume these are years 1, 2 and 3].

R.- We did so.

  1. Page 4, section a) how big were the seedlings? Were they container grown or bare root? If container grown, what volume and in what medium (which was then presumably transferred with them)? What fertilizer formula was applied, at what rate (per planting hole and translated to per ha).

R.- Please see response to a previous general comment.

  1. Page 4., Section 2.3, L. 4: confirm that you only measured trees with DBH > 15 cm for the Year 3 sampling. This suggests that there were probably many additional (smaller trees) found on these sites which are not included in the analysis, which really emphasizes only the most successful ones. This needs to be emphasized.

R.- Please see response to the last general comment above.

  1. Page 5, Lines 1 & 2: from where were these successional stage labels derived? [you quote sources for dispersal and pollination syndromes; please do the same here].

R.- We quoted the sources (reference 58).

  1. Page 5, 2nd para., L. 2; also 3rd para., L. 4: match the font of URLs with the rest of the text.

R.- We double checked all URL fonts for consistency.

  1. Page 5, Results [not Result].

R.- We did so (line 188).

  1. Section 3.1, L. 1: Instead of “surveyed” maybe say “observed” or “detected” or “identified”

R.- We did so (line 190).

  1. Page 6, L. 1 & L. 5 under the Table: Protium sp. and Cecropia sp. should have sp. without italics.

- same correction needed in Figure 3 and anywhere else in the manuscript.

R.- We did so.

  1. Page 6, last 2 lines: rephrase, dropping “were the most dispersed, i.e.”.

R.- We did so (line 212).

  1. Page 7, Section 3.3., L. 1: dispersal [not dispersion].

R.- We did so (line 222).

  1. Page 7, Sect. 3.3, L. 4: “widely spread” [so as not to confuse with seed dispersal syndrome].

R.- We did so (line 225).

  1. Page 8, Fig. 4, Panel c): Dispersal syndrome.

R.- We did so.

  1. Page 8, Lines 2-6 under Fig. 4: please match the number of decimal places in the means to that used for standard error (S.E.).

R.- We did so (lines 235-236).

  1. Page 8, L. 8: dispersal syndrome

R.- We did so (line 238).

  1. Page 9, Fig. 5: dispersal syndrome [both in the caption and in the panel a) label].

R.- We did so.

  1. Page 10, L. 2: re: diversity in the planted treatment sites – were these spp. volunteers on the planting sites, or are you referring solely to those planted? Or (ideally), can you provide the contribution of each origin?

R.- We rephrased as follows for clarification: “Only 11 species found in this study have been evaluated by the National Center for Plant Conservation and all of them were planted”.

  1. Page 10, Section 4.1, L. 5: rephrase to “…to what extent it pays to use active …”.

R.- We reworded as 'is justified to use' accordingly to another reviewer.

  1. Page 10, 2nd-last line of first paragraph in Section 4.1: “… or is similar to …”.

R.- We did so (line 297).

  1. Page 10, last line of first para. In Sect. 4.1: “…sites tend to be less disturbed”.

R.- We did so (line 298).

  1. Page 10, 2nd paragraph in Sect. 4, L. 4: biotic limitations [not limitation].

R.- We did so (line 303).

  1. Page 10, 2nd paragraph in Sect. 4, - Line 6: “indicates” [rather than “reflects”].

R.- We did so (line 305).

  1. Page 10, 2nd paragraph in Sect. 4, - Lines 10 & 11: “dispersal” [not dispersion, both instances].

R.- We did so (line 309-310).

  1. Page 11, L. 1: please hyphenate “species-rich family”.

R.- We did so (line 316).

  1. Page 11, Section 4.2, first para., L. 6: delete “the” before “central”.

R.- We did so (line 330).

  1. Page 11, Section 5 heading: Conclusions [not Conclusion – you clearly have several conclusions!].

R.- We did so (line 347).

  1. Page 11, Section 5, Line 8 under Conclusions: dispersal.

R.- We did so (line 354).

Response to Reviewer #2

Note: the line numbers by our responses to the comments refer to the clean version.

General comments

“An interesting and well presented study. While the key findings are not ground-breaking and have been found in other studies, I am unsure of similar work in the tropics so assume it has some novel factor because of this. I have suggested a number of text edits and/or clarifications for your consideration in the accompanying pdf (which includes supplementary file), primarily aimed at improving readability. Overall though a good manuscript, well done.”

R.- We appreciate the general positive comment about our ms.

Minor comments

Please note that we corrected the text throughout the ms as suggested by the reviewer in his/her annotated PDF; the changes are highlighted in the revised version. On top of these corrections, we accepted of his/her suggestions as follows.

Page 3, (2.2. Forest restoration techniques, 1ºparagraph): would be interesting to know where this soil was sourced?

R.- We explained this detail.

Page 11, (4. Discussion, 4.2. Proximity to seed sources accelerates vegetation recovery, 1ºparagraph): is 'high seed pressure' the correct term?

R.- We think so.

Finally, we wrote the legends of Fig. S1 and Table S1 and completed the author contributions, funding and statement of conflict of interest.

Reviewer 2 Report

Dear Authors

An interesting and well presented study. While the key findings are not ground-breaking and have been found in other studies, I am unsure of similar work in the tropics so assume it has some novel factor because of this. I have suggested a number of text edits and/or clarifications for your consideration in the accompanying pdf (which includes supplementary file), primarily aimed at improving readability. Overall though a good manuscript, well done.

Author Response

(The authors gave the same response as above.)
